# Aortic stenosis post-COVID-19: a mathematical model on waiting lists and mortality

Christian Philip Stickels [iD],[1] Ramesh Nadarajah [iD],[2,3,4] Chris P Gale,[2,3,4] Houyuan Jiang [iD],[5] Kieran J Sharkey [iD],[1] Ben Gibbison,[6] Nick Holliman [iD],[7] Sara Lombardo [iD],[8] Lars Schewe [iD],[9] Matteo Sommacal [iD],[10] Louise Sun [iD],[11,12] Jonathan Weir-McCall,[13,14] Katherine Cheema,[15] James H F Rudd,[16] Mamas Mamas [iD],[17] Feryal Erhun [iD][5]

For numbered affiliations see end of article.

**Correspondence to**
Dr Feryal Erhun;
f.erhun@jbs.cam.ac.uk

## ABSTRACT

**Objectives** To provide estimates for how different treatment pathways for the management of severe aortic stenosis (AS) may affect National Health Service (NHS) England waiting list duration and associated mortality.

**Design** We constructed a mathematical model of the excess waiting list and found the closed-form analytic solution to that model. From published data, we calculated estimates for how the strategies listed under Interventions may affect the time to clear the backlog of patients waiting for treatment and the associated waiting list mortality.

**Setting** The NHS in England.

**Participants** Estimated patients with AS in England.

**Interventions** (1) Increasing the capacity for the treatment of severe AS, (2) converting proportions of cases from surgery to transcatheter aortic valve implantation and (3) a combination of these two.

**Results** In a capacitated system, clearing the backlog by returning to pre-COVID-19 capacity is not possible. A conversion rate of 50% would clear the backlog within 666 (533–848) days with 1419 (597–2189) deaths while waiting during this time. A 20% capacity increase would require 535 (434–666) days, with an associated mortality of 1172 (466–1859). A combination of converting 40% cases and increasing capacity by 20% would clear the backlog within a year (343 (281–410) days) with 784 (292–1324) deaths while awaiting treatment.

**Conclusion** A strategy change to the management of severe AS is required to reduce the NHS backlog and waiting list deaths during the post-COVID-19 'recovery' period. However, plausible adaptations will still incur a substantial wait to treatment and many hundreds dying while waiting.

## STRENGTHS AND LIMITATIONS OF THIS STUDY

⇒ Our model provides a good basis from which to alleviate a time-critical health system problem when data gathering is likely to result in a greater number of deaths.

⇒ Offering transcatheter aortic valve implantation to some surgical aortic valve replacement patients in what might be considered suboptimal per-patient treatment in ideal conditions could result in better target population-based outcomes.

⇒ The assumption that the entire NHS can be modelled as a single entity with a single waiting list is a limitation of this study.

⇒ We recognise that the waiting numbers used in our study are only estimates because we do not know how many patients with AS died due to COVID-19 infection.

deferral of all but the most urgent interventional procedures and operations.[1 2]

Aortic stenosis (AS) is the most common form of valvular heart disease. Once stenosis is severe, symptoms follow and the prognosis is poor, with 50% mortality within 2 years of symptom onset.[3] Thus, timely treatment is of paramount importance. Surgical aortic valve replacement (SAVR) has historically been the default treatment strategy. However, transcatheter aortic valve implantation (TAVI) has recently emerged as an effective and increasingly used option across operative risk strata.[4–8]

There was a large decline in TAVI and SAVR procedural activity to treat severe AS during the COVID-19 pandemic.[9] Between the period March to November 2020, it is estimated that the decrease in activity accounted for 4989 (95% CI 4020 to 5959) patients in England with severe AS left untreated by TAVI or SAVR.[9] As we move into an era of 'living with' COVID-19, plans must urgently

## INTRODUCTION

The COVID-19 pandemic has led to the reorganisation of healthcare services to limit the transmission of the virus and deal with the sequelae of infection. This reorganisation had a detrimental effect on cardiovascular services, with reductions in hospitalisations for acute cardiovascular events and the

be put in place to best manage the additional waiting list burden for treatment of severe AS.[10]

In this study, we used mathematical methods to examine the extent to which additional capacity to provide treatment of severe AS should be created to clear the backlog and minimise deaths of people on the waiting list.

## METHODS
### Study population and assumptions

Data from the UK TAVR registry and National Institute for Cardiovascular Outcomes Research National Adult Cardiac Surgery Audit between 2017 and 2020 have previously been extracted to estimate an excess waiting list size ($W_0$) of 4989 (95% CI 4020 to 5959) patients with severe AS left untreated as of November 2020.[9] In the absence of contemporaneous data on waiting lists and SAVR and TAVI activity, we have taken this number as the excess backlog on which to model solutions. The incidence of AS has not increased over recent years.[11] Therefore, we assumed that the system was in a steady state before the COVID-19 pandemic and without loss of generality defined the steady-state waiting list to be zero. Additionally, we assumed that the normal rate of flow ($f$) of new patients into the waiting list for treatment of severe AS would be maintained on the commencement of additional operations. Thus, the extra capacity that we model is to clear the excess post-COVID-19 backlog.

We took 1 year mortality ($\mu$) after the onset of symptoms in severe AS to be 36% (95% CI 12% to 60%).[12] More recent studies have estimated the 1 year mortality to be 51%[5] and 55%,[13] but these included cohorts that were considered inappropriate for SAVR; thus, we considered these estimates unrepresentative of an unselected population with severe AS.[13] The routine capacity for treatment of severe AS was taken from the prepandemic period. In 2018/2019, the National Health Service (NHS) in England performed 7830 SAVR ($r_S^0 = 7830$) and 5197 TAVI ($r_T^0 = 5197$) procedures, for a total throughput of about 13 000 per year.[14]

### Modelling

Patients on the waiting list for treatment of severe AS were represented as a dynamical system (figure 1).

To this model, we introduced capacity in surplus to the 2018/2019 performance and called this capacity $T_e$ (further details are provided in online supplemental material). We assumed that the typical caseload for which the NHS in England can deal with continues; that is, we assumed that the system will return to prepandemic levels first using its baseline capabilities. The backlog accumulated during the pandemic is only reduced by treating patients with this extra capacity or by patient mortality before receiving treatment. We also considered patients in the backlog and patients new to the waiting list indistinguishable. Accordingly, the waiting list size represents the excess number of people seeking treatment who are unable to be treated immediately at any one time. We also assumed that other paths out of the waiting list (ie, patients seeking private treatment) would be so small in comparison to the uncertainty estimates as to be negligible on the results of our analysis.

These assumptions were brought together to give an estimated time (see online supplemental material for derivation) to clear the waiting list ($t_c$)

$$t_c = \frac{\ln\left(1 + \frac{W_0 \mu}{T_e}\right)}{\mu} \tag{1}$$

and associated mortality ($m(t_c)$)

$$m(t_c) = W_0 - T_e t_c. \tag{2}$$

Using equations (1) and (2), we predicted the length of time and associated mortality for different percentage increases in capacity. We assumed any capacity increase to be constant throughout the entire modelled period. For example, if we increased daily capacity by 5% this would result in, $T_e = \frac{r_S^0 + r_T^0}{365} * 5\% = 1.785$ extra procedures per day, across the whole of the NHS in England. We generated 10 000 random values for the 1-year mortality rate and initial waiting list length. We assumed that the uncertainty

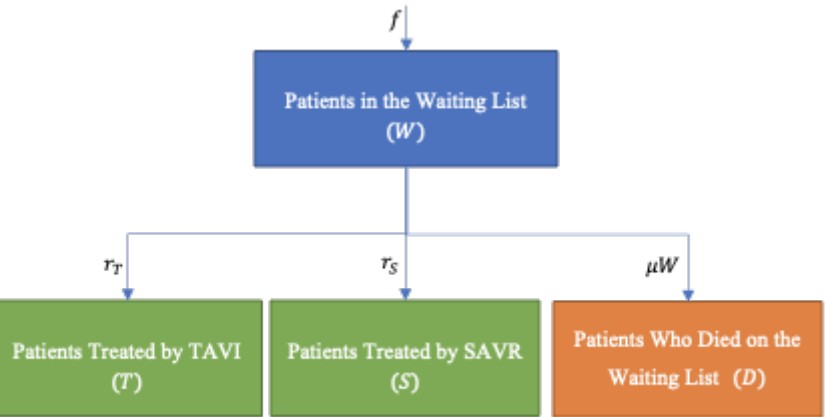

**Figure 1** Dynamical system model of the waiting list length. SAVR, surgical aortic valve replacement; TAVI, transcatheter aortic valve implantation.

in both variables was normally distributed. For every $T_e$, we present the mean and the 2.5 and 97.5 percentiles of the 10 000 simulations for time to clear the waiting list and the associated mortality. That is, we present the 95% reference range.[15]

### Interventions and outcomes

We investigated three types of capacity increase: (1) a general increase in the capacity to provide SAVR and TAVI, which could be facilitated by an increased number of procedures per list, additional lists and prioritisation of care pathways and staffing to treat severe AS; (2) extra capacity created by treating some patients with TAVI who would routinely have SAVR; (3) a combination of a general increase in capacity and the conversion of a proportion of cases from SAVR to TAVI. During the COVID-19 pandemic, TAVI was performed in patients usually referred for surgery, with no difference in short-term outcomes compared with historical reference groups.[16 17]

We assumed that the duration of a SAVR would routinely be between 2 and 4 hours and a TAVI between 1 and 2 hours.[18 19] As such, we assumed within the time for two SAVR operations, three TAVI could be performed instead.[20] Several clinical factors may favour SAVR over TAVI (including concomitant severe coronary artery disease, low STS score, bicuspid aortic valve etc); therefore, we assumed that, in the short term, no more than 50% of patients could be converted from SAVR to TAVI.[21] We also assumed that no more than 50% extra capacity could be created by other means (eg, extra lists, more procedures per list). We simulated two principal outcomes based on the creation of additional capacity ($T_e$): the time to clear the backlog (reduce to zero), and the mortality of patients within the excess backlog while on the waiting list to be treated.

We completed additional sensitivity analyses for how the conversion of SAVR to TAVI could affect the principal outcomes, including if three SAVR operations could be routinely completed in a day and four to five TAVI procedures per day (presuming increasing uptake of a minimalist TAVI approach without general anaesthesia enabling more rapid procedure time).[22]

### Patient and public involvement

Patients and the public were not involved in the conduct of this study.

## RESULTS

In the pre-COVID-19 period, the routine capacity for treatment of severe AS was set to cover the normal incident rate. That is, clearing the backlog by returning to pre-COVID-19 capacity is not possible. As a result, mortality on the excess waiting list at 1 year is estimated to be more than 1500, putting a strong emphasis on the need for change.

### Total additional capacity

Figure 2 provides simulations of the time to clear the excess backlog and the mortality of patients on the waiting list based on the amount of total additional capacity, $T_e$. With a 5% increase in the capacity to provide treatment of severe AS, we estimate it would take 1384 (1025–1994) days to clear the excess backlog, with 2526 (1355–3516) deaths. A 20% increase in total capacity would provide a benefit in clearing the excess backlog within 536 (434–666) days, with an estimate of 1173 (466–1859) deaths. As total capacity increased further, there was a diminishing return in clearing the backlog and avoiding associated mortality; the greater the capacity increase, the fewer lives are saved for every extra increase in capacity. Even if it was possible to double capacity, it was estimated that it may take 131 (126–137) days to clear the backlog and there would be 313 (118–494) deaths on the waiting list.

### The effect of converting SAVR to TAVI

The conversion of a proportion of cases from surgery to TAVI provides a modest improvement in estimates of

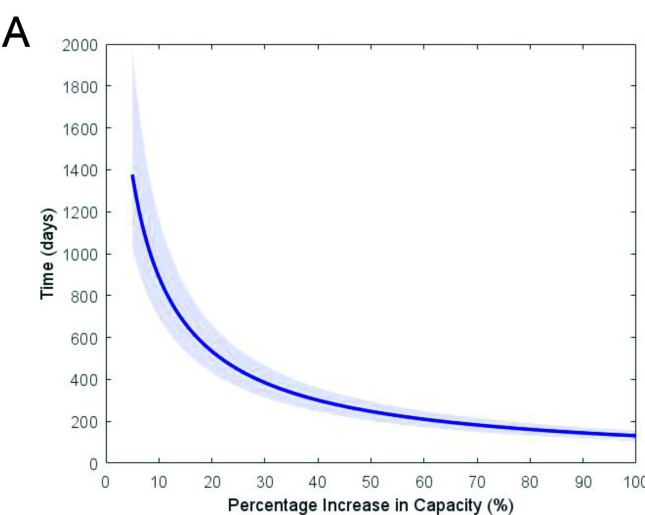
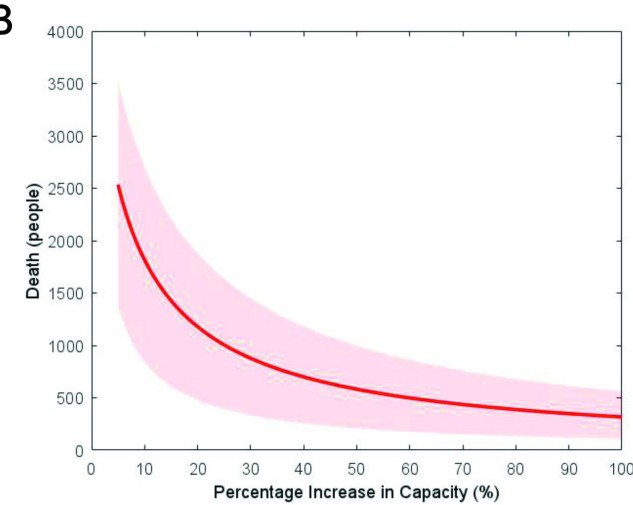

**Figure 2** Time to clear backlog (left) and the resulting deaths (right) with associated 95% reference range as a function of daily percentage increase in capacity, with uncertainty from mortality and the initial waiting list. The x-axis is truncated at 5% for visualisation and clarity.

time to clear the backlog and mortality on the waiting list. With the conversion of 30% of SAVR operations to TAVI procedures, without the creation of additional capacity in the system, we estimated that it would take 975 (741–1284) days to clear the backlog and there would be 1914 (923–2809) deaths on the waiting list. Even with a conversion of 50% of SAVR operations to TAVI procedures, the estimated backlog would be cleared within 666 (533–848) days with 1419 (597–2189) deaths. For the highest conversion ratio that we considered (2:4), at a 50% rate of conversion, we estimated the backlog to be cleared in 384 (330–462) days with 871 (314–1426) deaths. While this result is improved, we consider a 2:4 conversion ratio the highest reasonable ratio in the short term, and is unlikely to be achieved at every centre immediately. It is also worth noting that even if this was achieved, the backlog would still take over a year to clear.

**Combining conversion of SAVR to TAVI and additional capacity**
Figure 3A,B demonstrates the range of possibilities in creating extra capacity. Each line demonstrates a range of intervention strategies that provide the same result. For example, to reduce the mean predicted deaths to 1000 (red line figure 3B), centres could increase capacity to provide an extra 25% procedures per week at the same mix as prepandemic, or they could convert 50% of SAVR operations to TAVI and increase their capacity by 8.7% at that mix. Figure 3C,D represents how the combinations of interventions to increase capacity within the system alongside the conversion of SAVR to TAVI would impact the time to clear the backlog and on the associated mortality of waiting. Mortality on the waiting list is less responsive to our modelled interventions than the time to clear the backlog (the darker coloured regions of figure 3D make up a greater proportion of the estimates

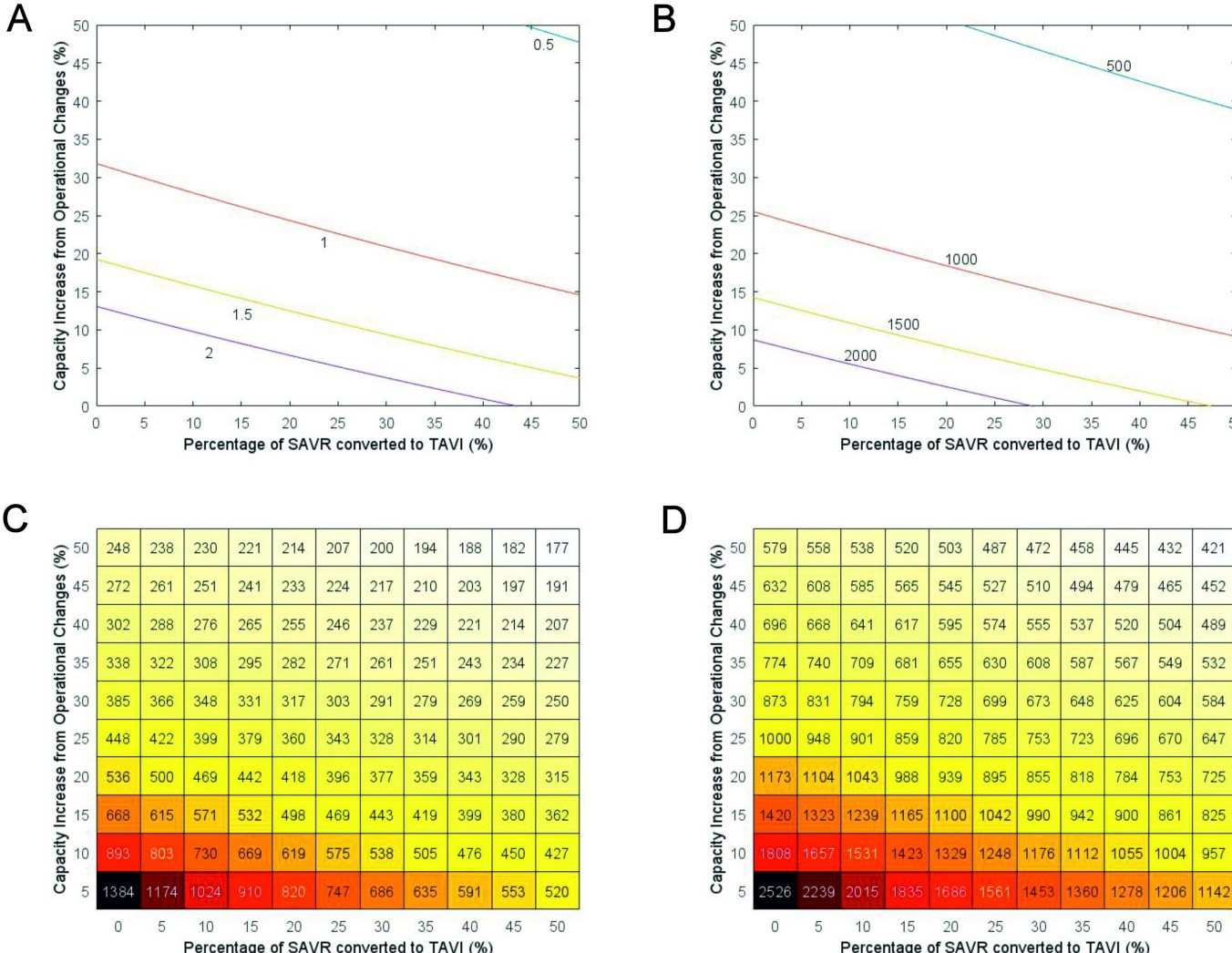

**Figure 3** Mean time to clear backlog (left) and the resulting deaths (right) as a function of daily percentage increase in capacity (y-axis) and percentage of surgical aortic valve replacement (SAVR) converted to transcatheter aortic valve implantation (TAVI) (x-axis) (presented in two different forms). (A) Isoclines of constant mean clearance-time going from half a year (blue) to 2 years (purple) in half-year increments. (B) Isoclines of constant mean mortality after clearing the backlog from 500 people (blue) to 2000 (purple) in 500-person increments. (C) Heatmap of different combinations of conversion and daily capacity increases and how long the backlog would take to clear on average, in days. (D) Heatmap of different combinations of conversion and daily capacity increases and how many people would die, on average.

than those of figure 3C). Increasing capacity within the system alongside converting a proportion of SAVR cases to TAVI provides the greatest estimated benefit in clearing the backlog and avoiding associated mortality. A combination that would result in the clearance of the backlog within a year might be of interest for decision-makers. With the conversion of 40% of SAVR operations to TAVI and the creation of an additional 20% capacity, we estimated that the backlog would be cleared in just under a year—343 days (281–410) with 784 (292–1324) deaths before treatment.

Sensitivity analyses where the number of TAVI procedures that could be completed within the same time as SAVR was altered (TAVI to SAVR: 4 to 3, 4 to 2, 5 to 3) support these findings (online supplemental material figures S1–S3). Furthermore, sensitivity analyses show that with the best-in-class practices (TAVI to SAVR: 4 to 2), even a more modest combination (a conversion of 35% and creation of an additional 10% capacity) may be enough to clear the backlog within a year.

## DISCUSSION

In this study, using dynamical system modelling, we provide estimates for how changes to treatment pathways for patients with severe AS may affect the time taken to clear the backlog and minimise mortality on the waiting list in the NHS of England. Without providing at least 20% total additional capacity for the interventional treatment of AS, we estimated there would be more than 1000 deaths on the waiting list over a period of nearly 1.5 years. A conversion of cases from SAVR to TAVI would expedite the clearance of the backlog, but even converting half the cases to TAVI would still result in over 1400 deaths over a period of almost 2 years. A combination of converting 40% of cases usually planned for SAVR to TAVI and creating 20% additional capacity for procedures (through measures such as extra lists) would clear the excess backlog within 1 year, with 784 deaths.

Our study has several strengths. First, in a time-critical clinical situation of many unknowns, our use of novel mathematical models provides plausible estimates on which to base health services planning, and provides an exemplar that may be used in service delivery in other conditions in the postpandemic landscape. Given the high event rate among this population, waiting for more contemporary data to be collected may well not provide enough time to institute system changes to prevent deaths. Second, we also provide specific estimates for how the conversion of cases to TAVI from surgery may affect waiting lists and associated mortality, which can inform local multi-disciplinary team (MDT) discussions. Third, our model can act as a basis for a clinical and cost–benefit analysis to evaluate different ways to increase capacity and define the optimal strategy at each centre. For each centre, the most effective combination of converting SAVR to TAVI and provision or prioritisation of treatment of severe AS can be generated.

We recognise the limitations inherent in modelling a complex situation. First, we represent the NHS in England as a single entity. As such, we implicitly assume that population and capacity are distributed evenly throughout the country by treatment centre capacity. If the distribution of waiting list patients deviates significantly from the distribution of treatment centres weighted by capacity, the time it would take to clear the waiting list, and thus the mortality rate would be higher. Second, we have not attempted to calculate how many patients with AS may have died in the COVID-19 pandemic, which could have reduced the numbers of deaths on the waiting list and the duration of the waiting list because of an underestimation of 'abandonment' from the model. Third, our assumed mortality rate may differ at a centre-level due to prioritising clinically more vulnerable patients on the waiting list. Fourth, a centre-level analysis could account for the different practices in each treatment centre and identify strategies that work best for each centre. Fifth, our estimates from converting cases from SAVR to TAVI do not include postprocedural factors such as the requirement for intensive care capacity, hospital stay and further procedures because these rely on multiple centre-specific factors. Finally, it has been shown that rapid growth in the demand for TAVI can overwhelm current capacity,[23] which may lead to prolonged wait times and subsequent adverse outcomes while patients are on the waitlist. Therefore, a demand model that captures the changes of demand for TAVI and SAVR would be a helpful future direction of analysis.

A previous study used a mathematical model to quantify the cumulative cardiac surgical backlog (including coronary artery bypass grafting surgery, valve replacement and transcatheter aortic and mitral valve replacements) in two centres based on the projected pandemic duration in the USA.[24] The authors used simple mathematical models to predict the time required to clear the backlog depending on increased operating capacity. However, the authors did not consider mortality, which we have as it is of critical importance to patients and when planning services.

The results of our study highlight concerns pertaining to the deferral of non-emergency treatment for severe AS during the 'recovery period' of COVID-19. Severe AS is a progressive condition with valve replacement the only available treatment improving prognosis.[25] On a local, regional and national scale, healthcare systems will need to examine capacity, set priorities and plan for adequate capacity to manage the backlog of patients with severe AS. The response will be complicated by prior exhaustion of human resources from the pandemic and competition with other specialities, which will also have backlogs.[26]

Nonetheless, planning should prioritise patients at the highest risk from a deferral of treatment. Mortality on the waiting list for AS has been reported to be as high as 14%.[27] Furthermore, patients awaiting structural procedures deferred due to the pandemic have been found to have significantly higher mortality rates compared with those with stable coronary artery disease.[28] Prioritising capacity for treatment of patients with severe AS may mean reduced capacity for other procedures. Providing

20% extra capacity for TAVI and SAVR may only require the one or two additional procedures each week per centre at the expense of other procedures, as many centres only conduct TAVI procedures on between 2 and 3 days per week.[22] This interaction will require collaborative decision-making on a local level accepting that these are difficult, imperfect times. We also show that the conversion of a proportion of cases that would usually be managed by SAVR to TAVI can help expedite treatment and reduce mortality on the waiting list. During the pandemic, TAVI procedures were performed in patients usually referred for surgery with no apparent difference in short-term outcomes;[16 17] and data continue to emerge for longer term efficacy and safety of TAVI across operative risk strata.[29 30] Recent European guidelines suggest that TAVI would be a preferable option for patients over 75 years of age compared with SAVR.[21]

To help planning, we provide an app (https://github.com/Christian-P-Stickels/AS_Waitinglist_data) to explore the impact of alterations in capacity and treatment pathways on waiting lists and mortality-related to severe AS at a local, regional and national level (online supplemental material).

## CONCLUSIONS

In this study, we found that without a combination of increased capacity for treatment of patients with severe AS and an expansion in the use of TAVI, there would be many potentially avoidable deaths during the post-COVID-19 recovery period. Our study findings and accompanying app may help inform the planning of cardiac services.

**Author affiliations**
¹Department of Mathematical Sciences, University of Liverpool, Liverpool, UK
²Leeds Institute for Data Analytics, University of Leeds, Leeds, UK
³Leeds Institute of Cardiovascular and Metabolic Medicine, University of Leeds, Leeds, UK
⁴Department of Cardiology, Leeds Teaching Hospitals NHS Trust, Leeds, UK
⁵Judge Business School, University of Cambridge, Cambridge, UK
⁶Cardiac Anaesthesia and Intensive Care, Bristol Medical School, Bristol, UK
⁷Department of Informatics, King's College London, London, UK
⁸Department of Mathematical Sciences, Loughborough University, Loughborough, UK
⁹School of Mathematics and Maxwell Institute for Mathematical Sciences, University of Edinburgh, Edinburgh, UK
¹⁰Department of Mathematics, Physics and Electrical Engineering, Northumbria University, Newcastle upon Tyne, UK
¹¹Division of Cardiac Anesthesiology, University of Ottawa Heart Institute, Ottawa, Ontario, Canada
¹²Cardiovascular Research Program, Institute for Clinical Evaluative Sciences, Toronto, Ontario, Canada
¹³Department of Radiology, University of Cambridge, Cambridge, UK
¹⁴Department of Radiology, Royal Papworth Hospital, Cambridge, UK
¹⁵Health Intelligence, British Heart Foundation, London, UK
¹⁶Department of Medicine, University of Cambridge, Cambridge, UK
¹⁷Keele Cardiovascular Research Group, Keele University, Keele, UK

**Acknowledgements** We want to thank all the participants of the V-KEMS Study Group on 'Modelling Solutions to the Impact of COVID-19 on Cardiovascular Waiting Lists' that took place on February 2-4, 2021, for thought-provoking discussions. Our special thanks to Clare Merritt (Newton Gateway to Mathematics), whose help extended beyond the workshop and was crucial in completing this work, and to Alan Champneys who brought the group together in the first place. BG is supported by the NIHR Bristol Biomedical Research Centre at the University of Bristol and University Hospitals Bristol and Weston NHS Foundation Trust. JHFR is part-supported by the NIHR Cambridge Biomedical Research Centre, the British Heart Foundation, HEFCE, the EPSRC Cambridge Centre for Mathematics of Information in Healthcare and the Wellcome Trust.

**Contributors** MM proposed the initial workshop and designed the research question. MM, CPG, RN, BG and JHFR all helped to run said workshop as clinical experts. All members but KC and FE were involved in conceptualisation in the initial workshop. CPS, HJ, KJS and FE designed the model with clinical guidance from MM, CPG, RN, BG and JHFR. CPS performed data analysis. CPS, RN and FE drafted the initial manuscript. MM, CPG, BG, JHFR, NH, SL, LaSc, MS, LoSu, JW-M, KC provided critical interpretation and revision of the manuscript. All authors approved the final manuscript. FE acts as the guarantor for the overall content.

**Funding** This study was part funded by EPSRC Cambridge Centre for Mathematics of Information in Healthcare, grant number EP/T017961/1. None of the study funding sources had an impact on the design, data analysis, writing of or decision to publish this paper.

**Competing interests** BG acknowledges grants not related to this project from the David Telling Charitable Trust, and the Biotechnology and Biological Sciences Research Council, he additionally declared Associate Editorship of Anesthesia Journal, and being the chair DMSC for the COPIA Trial. All other authors confirm that they have no competing interests to declare.

**Patient and public involvement** Patients and/or the public were not involved in the design, or conduct, or reporting, or dissemination plans of this research.

**Patient consent for publication** Not applicable.

**Provenance and peer review** Not commissioned; externally peer reviewed.

**Data availability statement** Data sharing not applicable as no datasets generated and/or analysed for this study.

**ORCID iDs**
Christian Philip Stickels http://orcid.org/0000-0002-5286-9379
Ramesh Nadarajah http://orcid.org/0000-0001-9895-9356
Houyuan Jiang http://orcid.org/0000-0002-2466-6929
Kieran J Sharkey http://orcid.org/0000-0002-7210-9246
Nick Holliman http://orcid.org/0000-0003-4418-4908
Sara Lombardo http://orcid.org/0000-0003-3545-163X
Lars Schewe http://orcid.org/0000-0002-3778-262X
Matteo Sommacal http://orcid.org/0000-0003-2820-2117
Louise Sun http://orcid.org/0000-0003-3381-3115
Mamas Mamas http://orcid.org/0000-0001-9241-8890
Feryal Erhun http://orcid.org/0000-0003-0339-7085

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
