## [Reviewer comments · BMJ Open]

ARTICLE DETAILS

TITLE (PROVISIONAL)	Aortic stenosis post-COVID-19: A mathematical model on waiting lists and mortality
AUTHORS	Stickels, Christian; Nadarajah, Ramesh; Gale, Chris; Jiang, Houyuan; Sharkey, Kieran J; Gibbison, Ben; Holliman, Nick; Lombardo, Sara; Schewe, Lars; Sommacal, Matteo; Sun, Louise; Weir-McCall, Jonathan; Cheema, Katherine; Rudd, James H F; Mamas, Mamas; Erhun, Feryal

VERSION 1 – REVIEW

REVIEWER	Segev, Amit Sheba Medical Center at Tel Hashomer
REVIEW RETURNED	08-Feb-2022

GENERAL COMMENTS	This is a timely and excellent paper showing the urgent steps required to maintain elective structural interventions during COVID19 outbreak. Only 1 comment: Authors show that conversion from SAVR to TAVI resulted in only modest effect on clearing the waiting list for AS treatment. If TAVI performed in a minimalist approach, e.g. local anesthesia, direct admission to cardiology rather to CCU, early home monitoring with auto-triggered loop recorder and more, would it change dramatically the ability to perform more percutaneous procedures? Can the authors include it in the mathematical model?
--

REVIEWER	Shoukat, Affan Yale University
REVIEW RETURNED	07-Mar-2022

GENERAL COMMENTS	I have reviewed the manuscript assigned. In this paper, the authors conduct a modelling study to clear the backlog of AS cases currently in the UK. While simple in its nature, I found this paper technically sound, easy to read, and most importantly, presents important insights for NHS decision-makers. I recommend this manuscript for publication, though have a few minor comments. I would be interested to hear the authors remarks on these comments. 1. Can the results be discussed within the context of NHS limitations? Is it feasible to increase capacity by 20%? How hard is it to convert between the two types of interventions described in the paper.
--

	2. The authors assume no increasing incidence and reference a paper published in 2015. However, I find that study to be specific to the Swedish population. Are the two populations (UK and Sweden) demographically similar for this assumption to hold? 3. The authors state that there are roughly 13,000 procedures done per year. It's not clear to me how much did it get reduced to during COVID19? When discussing results specifically for increase in capacity, do the authors assume return to pre-pandemic levels first (i.e. 13,000 procedures)? Also, should time play a significant role here? If the number of procedures have dropped significantly during the pandemic, might it be a couple of months and/or years to ramp capacity back to normal? If so, this should be discussed. 4. A quick follow up question. Do the 13,000 procedures clear all AS cases in that year or has the backlog been consistently increasing over time? 5. Is the uncertainty in the results only coming from the 10,000 random numbers for mortality and initial waiting list estimate? Could other variables also be randomized? Are the authors really calculating a confidence interval or might it be better described as a credible interval. 6. While discussed as a limitation, a larger impact could be achieved if considering more heterogeneity (i.e., waiting lists for specific locations instead of the entire region). 7. Lastly, it would be interesting to see a cost-effectiveness analysis of the two interventions described in this study.
--	--

VERSION 1 – AUTHOR RESPONSE

Reviewer: 1

Dr. Amit Segev, Sheba Medical Center at Tel Hashomer

This is a timely and excellent paper showing the urgent steps required to maintain elective structural interventions during COVID19 outbreak.

Dr Segev, many thanks for your kind words regarding our paper.

Authors show that conversion from SAVR to TAVI resulted in only modest effect on clearing the waiting list for AS treatment. If TAVI performed in a minimalist approach, e.g. local anesthesia, direct admission to cardiology rather to CCU, early home monitoring with auto-triggered loop recorder and more, would it change dramatically the ability to perform more percutaneous procedures? Can the authors include it in the mathematical model?

Author reply

We thank you for this excellent point. The SAVR to TAVI conversion rate we used in the main body of the paper is 2 to 3. However, we did a sensitivity analysis on this conversion rate. In the Supplements, we present the results for higher conversions rates (3:5 and 2:4) to take into account the shorter procedure time involved if a minimalist approach to TAVI was taken. We now included a brief discussion on these sensitivity results (conversion only) to the main body of the paper.

Please note that we cannot credibly go above the 2:4 conversion rate. Without considering other factors (such as workforce, bed availability in a given day, etc.), such an analysis would not be robust. We deliberately did not model these other factors as they would be related to the infrastructure and the particular practices at each centre. We acknowledge that centres may improve their practices over time through learning, process alignments, etc. However, we do not see these improvements taking effect in the next 12-18 months.

Manuscript change

For the highest conversion ratio that we considered (2:4), at a 50% rate of conversion, we estimated the backlog to be cleared in 384 (330–462) days with 871 (314–1426) deaths. Whilst this result is improved, we consider a 2:4 conversion ratio the highest reasonable ratio in the short-term, and is unlikely to be achieved at every centre immediately. It is also worth noting that even if this was achieved, the backlog would still take over a year to clear.

Reviewer: 2

Dr. Affan Shoukat, Yale University

I have reviewed the manuscript assigned. In this paper, the authors conduct a modelling study to clear the backlog of AS cases currently in the UK. While simple in its nature, I found this paper technically sound, easy to read, and most importantly, presents important insights for NHS decision-makers. I recommend this manuscript for publication, though have a few minor comments. I would be interested to hear the authors remarks on these comments.

Dr Shoukat, we are glad you thought we provided important and sound insights to NHS. We appreciate your time and suggestions. Please see below for our response to your comments.

1. Can the results be discussed within the context of NHS limitations? Is it feasible to increase capacity by 20%? How hard is it to convert between the two types of interventions described in the paper.

Author reply

Two of the main limitations on the provision of intervention for aortic stenosis are commissioning and competing capacity needs in catheterisation laboratories (PMID 33767000). In the case of TAVI in 2019 in the UK, only 78 procedures per million population (pmp) were undertaken, compared with a European average of 141 pmp. Many centres may only perform TAVI procedures 2-3 days per week. To provide a 20% increase in structural aortic valve interventions may only require the addition of 1 list per week for SAVR and TAVI. This change is manageable if departments can agree on appropriate prioritisation of cardiac procedures, especially given the mounting evidence of high mortality amongst patients waiting for structural heart disease interventions that were deferred during the COVID-19 pandemic (PMID: 33336506). We hope that disseminating the results presented in this paper will allow for discussions within centres around prioritisation of structural heart interventions in the catheterisation lab, now that other invasive procedures (coronary angiography) can increasingly be replaced by non-invasive coronary CT as outpatient procedures. Furthermore, this data may influence commissioners to provide sufficient funds for more procedures to avoid unnecessary loss of life.

It is possible for some patients who may have been initially treated with SAVR to be converted to TAVI, particularly when randomised controlled trials have shown similar outcomes in selected intermediate and low risk patient groups undergoing TAVR or SAVR. During the pandemic, evidence

has emerged of NHS centres moving from 45% of patients being treated with TAVI to 73% (PMID 33565703). Furthermore, as the most recent European Society of Guidelines have been updated to advocate the preferential treatment of individuals over the age of 75 years with TAVI, we expect a significant proportion of previously treated cases with SAVR will now be treated with TAVI as part of standard clinical practice.

Manuscript change

Providing 20% of extra capacity for TAVI and SAVR may require only the addition of one extra list per week at the expense of other procedures, as many NHS centres only conduct TAVI procedures on between two to three days per week.

2. The authors assume no increasing incidence and reference a paper published in 2015. However, I find that study to be specific to the Swedish population. Are the two populations (UK and Sweden) demographically similar for this assumption to hold?

Author reply

Life expectancy and population age structure are broadly similar in Sweden and the UK.^{1,2,3,4} There is an exponential increase in prevalence in aortic stenosis with age, and calcific aortic valve disease is the predominant pathology leading to severe aortic stenosis across the USA and Europe. Thus, we believe it is reasonable to use incidence trends from the Swedish population as a proxy for the UK. In our literature search, we have not found more recent literature that refutes this assumption.

¹<https://www.nationmaster.com/country-info/compare/Sweden/United-Kingdom/Health>

²<https://www.ons.gov.uk/peoplepopulationandcommunity/populationandmigration/populationestimates/articles/ukpopulationpyramidinteractive/2020-01-08>

³<https://www.statista.com/statistics/521717/sweden-population-by-age/>

⁴<https://www.esccardio.org/Journals/E-Journal-of-Cardiology-Practice/Volume-18/epidemiology-of-aortic-valve-stenosis-as-and-of-aortic-valve-incompetence-ai>

3. The authors state that there are roughly 13,000 procedures done per year. It's not clear to me how much did it get reduced to during COVID19?

Author reply

During the first wave of the pandemic, there was indeed a step decrease in the number of procedures performed (Martin et al. 2021, PMID: 34003671):

“For the first few months after lockdown, the estimated difference (95% CI) in the number of TAVR cases per month compared with those expected based on historic trends was -2 (-40 to 35) in March 2020, -229 (-264 to -193) in April 2020, -191 (-229 to -154) in May 2020, and -129 (-166 to -92) in June 2020 ... The estimated decrease in isolated AVR activity was -171 (-201 to -140), -231 (-257 to -205), -177 (-205 to -148) and -96 (-124 to -69), across March to June 2020, respectively.”

This paper suggested that during the decline and recovery phases, the decrease in activity accounted for 4,989 patients in England with severe AS left untreated by TAVI or SAVR. We used this as the excess waiting list to be treated when we modelled the interventions without more contemporaneous data.

When discussing results specifically for increase in capacity, do the authors assume return to pre-pandemic levels first (i.e. 13,000 procedures)?

Author reply

We choose to go back to the pre-pandemic levels in our analysis first. We chose this for two reasons: (1) this time frame is usually used as the baseline within NHS to have the recovery effort discussions; (2) this was the most evidenced baseline available in the absence of contemporaneous nationwide data. To make this more apparent in the paper, we updated the related paragraph in Methods:

Manuscript change

We assumed that the typical caseload for which the NHS in England can deal with continues; i.e., we assumed that the system will return to pre-pandemic levels first using its baseline capabilities. The backlog accumulated during the pandemic is only reduced by treating patients with this extra capacity or by patient mortality before receiving treatment.

Also, should time play a significant role here? If the number of procedures have dropped significantly during the pandemic, might it be a couple of months and/or years to ramp capacity back to normal? If so, this should be discussed.

Author reply

We agree with the reviewer. Previous work has shown that whilst there was partial recovery after lockdown for TAVR and SAVR procedures across the NHS, these did not return back to expected levels several months post lockdown (Martin et al. 2021, PMID: 34003671).

4. A quick follow up question. Do the 13,000 procedures clear all AS cases in that year or has the backlog been consistently increasing over time?

Author reply

In our analysis, we compare the situation with the status quo. That is, extra procedures will need to be performed in addition to the 13,000 to make up for the deficit that accrued during the pandemic. Our results do not suffer from this assumption due to how the equations are written (i.e., incremental rather than total).

There was a backlog in the NHS before the pandemic, with oscillating waiting lists at different centres. Fixing those structural problems is beyond the scope of our analysis. We believe studying individual centres would be a better approach to solving such problems, which can form part of future work.

5. Is the uncertainty in the results only coming from the 10,000 random numbers for mortality and initial waiting list estimate? Could other variables also be randomized?

Author reply

In theory, all our inputs can have randomness. However, we have concentrated on mortality and initial waiting list uncertainty because these two variables are exogenous to our study system. Therefore, understanding their variability is important as the decision-makers may not be able to influence these numbers.

Although the conversion ratio can be uncertain, it would be quite difficult to capture this meaningfully for two reasons: (1) It is unclear to us what a realistic distribution of the conversion ratio would be. (2) It is also unclear to us what this randomness would mean at a system level. We, however, note that one way to get an initial glimpse of potential uncertainty in conversion ratio is to compare the sensitivity results in the Supplements for different conversion ratios.

Are the authors really calculating a confidence interval or might it be better described as a credible interval?

Author reply

Many thanks for raising the issue with the credible vs. confidence intervals. We decided to follow a different path, and use neither. This decision is based on the guidance BMJ provides to the authors (<https://www.bmj.com/about-bmj/resources-readers/publications/statistics-square-one/4-statements-probability-and-confiden>). In short, we are presenting the reference ranges. We now provide a short explanation of the approach in the paper. Please see the manuscript change below.

Manuscript change

For every t , we present the mean and the 2.5 and 97.5 percentiles of the 10,000 simulations for time to clear the waiting list and the associated mortality. That is, we present the 95% reference range.

6. While discussed as a limitation, a larger impact could be achieved if considering more heterogeneity (i.e., waiting lists for specific locations instead of the entire region).

Author reply

We may indeed be underestimating the time to clear the waiting lists by ignoring the heterogeneity. Although at the beginning of the study we wanted to study this heterogeneity, we had to concede on this for two reasons:

- A. It is difficult to get data for all individual centres. Data availability is a challenge during regular times, yet it is especially taxing during the pandemic.
- B. Concentrating on a few select centres would not be scalable and may create resistance in contemplating our suggestions.

Therefore, we believe this focus is beyond the scope of the current paper and would benefit from further development as a separate study.

7. Lastly, it would be interesting to see a cost-effectiveness analysis of the two interventions described in this study.

Author reply

This is a good point. However, we believe it is beyond the scope of the current paper.

Our analysis does not aim to targeting a structural change in the NHS, but rather potential solutions that may be considered to address the current situation.

In addition, data (un)availability is of concern for a cost-effectiveness analysis. There are multiple ongoing studies on the effectiveness of TAVI for younger patients (NCT02825134, NCT03112980);

the results of these studies will be instrumental in a cost-effectiveness analysis. However, the current crisis does not allow us to wait for these results and collect additional, centre-specific data on cost-effectiveness.

VERSION 2 – REVIEW

REVIEWER	Segev, Amit Sheba Medical Center at Tel Hashomer
REVIEW RETURNED	17-May-2022
GENERAL COMMENTS	Authors have improved the manuscript significantly, which is now suitable for publication